



# On the role of operational dynamics in biogeochemical efficiency of a soil aquifer treatment system

Shany Ben Moshe[1], Noam Weisbrod[2], Felix Barquero[3], Jana Sallwey[3], Ofri Orgad[2], and Alex Furman[1]

[1]Technion - Israel Institute of Technology, Civil and Environmental Engineering, Haifa 32000, Israel
[2]The Zuckerberg Institute for Water Research, Blaustein Institutes for Desert Research, Ben Gurion University of the Negev, Israel
[3]Institute for Groundwater Management, Technische Universität Dresden, Dresden, Germany

**Correspondence:** Shany Ben Moshe (Benmoshe.shany@gmail.com)

**Abstract.** Sustainable irrigation with treated wastewater (TWW) is a promising solution for water scarcity in arid and semi-arid regions. Soil aquifer treatment (SAT) provides a solution for both the need for tertiary treatment and seasonal storage of wastewater. Stresses over land use and the need to control the obtained water quality makes the optimization of SAT of great importance. This study looks into the influence of SAT systems' operational dynamics (i.e. flooding and drying periods) as well as some aspects of the inflow biochemical composition on their bio-geo-chemical state and the ultimate outflow quality. A series of four long-column experiments was conducted, aiming to examine the effect of different flooding/drying period ratios on dissolved oxygen (DO) concentrations, oxidation-reduction potential (ORP) and outflow composition. Flooding periods were kept constant at 60 minutes for all experiments while drying periods (DP) were 2.5 and 4 times the duration of the flooding periods. Our results show that the longer DP had a significant advantage over the shorter periods in terms of DO concentrations and ORP in the upper parts of the column as well as in the deeper parts, which indicates that larger volumes of the profile were able to maintain aerobic conditions. This advantage was evident also in outflow composition analyses that showed significantly lower concentrations of DOC, TKN and ammonium in the outflow for the longer DP. Comparing experimental ORP values in response to different DP to field measurements obtained in one of the SAT ponds of the SHAFDAN, Israel, we found that despite the major scale differences between the experimental 1D system and the field 3D conditions, ORP trends in response to changes in DP, qualitatively match. We conclude that longer DP not only ensure oxidizing conditions close to the surface, but also enlarge the active (oxidizing) region of the SAT. While those results still need to be verified in full scale, they suggest that SAT can be treated as a pseudo-reactor that to a great extent could be manipulated hydraulically to achieve the desired water quality while increasing the recharge volumes.



# 1 Introduction

Water shortage in arid and semi-arid regions leads to great difficulties sustaining local agriculture which have many economical as well as societal and environmental implications (Garcia et al., 2014). The use of treated wastewater (TWW) for irrigation is widely accepted as one of the means to reduce agricultural water scarcity (Negewo et al., 2011). While biological treatments like activated sludge are highly efficient, their product usually does not meet regulatory standards for unrestricted crop irrigation (Tanji et al., 1997) or poses a sustainability question mark on the practiceAssouline et al. (2016). For example, in the Israeli

Dan region wastewater treatment plant (the SHAFDAN site), following the activated sludge secondary treatment, TWW have low to moderate organic load. Dissolved Organic Carbon (DOC) is $\sim 900 \mu M$, ammonium is $\sim 300\ \mu M$ and organic nitrogen is $\sim 90\ \mu M$ (Icekson et al., 2011). However, for the TWW to meet regulatory standards for unlimited irrigation, further water quality enhancement is required (Shuval et al., 1986). Soil aquifer treatment (SAT) may be the supplementary treatment component for conventional (secondary) WW treatment, that is needed to meet regulations and sustainability -these systems

involve clusters of infiltration ponds through which TWW are infiltrated through the vadose zone, into the aquifer in cycles of flooding and drying. In the ponds and subsequently in the unsaturated zone, residual dissolved organic carbon (DOC) and other nutrients (like organic and inorganic nitrogen species) are involved in bio-geo-chemical processes (such as adsorption to the soil minerals, consumption by bacteria etc.) which result in further decrease in the TWW's organic load and overall improved chemical composition (Bouwer et al., 1991; Amy et al., 2007).

In Israel, $\sim 150$ million $m^3$ of wastewater (WW) are treated each year in the SHAFDAN facility (Icekson et al., 2011). After the SAT process, the TWW are transported to the south of Israel and are used by farmers for crop irrigation) (Idelovitch et al., 2003). With the constant raise in population, SAT sites are often not able to treat all the WW they receive, which results in conventionally treated WW being directed to the local streams or the sea. Clearly, the SAT component is the bottleneck for full utilization of TWW. The SAT mechanism relies on the various bio-geo-chemical processes that take place during

TWW infiltration. These processes begin even before the TWW reach the unsaturated zone- Goren et al. (2014) described the variability in Carbon and nitrogen species through the hours of the day and in different seasons in a field study conducted in the SHAFDAN site (Goren et al., 2014). They found that the chemical composition of the TWW in the infiltration ponds responded to the day and night cycles. For example, during daytime, dissolved oxygen (DO) concentrations increase due to photosynthesis, reaching a maximum in the late afternoon. As a result, TWW that are infiltrated during the day are significantly

more oxidized compared to TWW that are infiltrated during the night, affecting the redox state of the soil profile and thus impact rates of oxidation reactions. Once the TWW reach the unsaturated zone, oxygen concentrations become a limiting factor that affects the efficiency of some key biogeochemical processes that are crucial for the enhancement of the TWW quality. The role of oxygen as a limiting factor in SAT systems is especially prominent in the deeper areas of the vadose zone where natural aeration during the drying periods (DP) is limited. DOC and nitrogen species' degradation during infiltration depend

heavily on processes like aerobic bacteria respiration and nitrification for which DO concentrations are of great importance (Goren et al., 2014). Mienis et al. (2018) studied nitrogen behavior under the SHAFDAN's infiltration ponds through 40 years of operation (Mienis et al., 2018). In their study, concentrations of ammonium, Total Kjeldahl Nitrogen (TKN) and oxidized





nitrogen species ($NO_2$, $NO_3^-$, $NO$) were tracked, using data from two observation wells, screening different depths of the vadose zone. They found that removal of up to $\sim 75\,\%$ of the total nitrogen occurred in the upper parts of the soil profile (up to $\sim 70cm$ below ground surface) while the deeper parts had a smaller contribution. This observation led them to the conclusion that inflow concentrations of more than 10 mg/L of ammonium will result in a decrease in reclaimed water quality due to ammonium and organic nitrogen leakage into the aquifer. Previous research regarding oxygen behavior in the vadose zone during wetting and drying cycles (Kim et al., 2004; Dutta et al., 2015) emphasize the role of the site's operational dynamics in the reclaimed water quality. During the flooding periods (FP), water content (WC) in the soil profile below the infiltration pond gradually increases (Arye et al., 2011), limiting the scope of diffusive and advective aeration which is crucial for the replacement of the DO used by bacteria. Hence, DP play a major role in the overall biogeochemical dynamics of SAT systems (Miller et al., 2006). Looking at oxygen dynamics during wetting and drying cycles in a 1-meter sand column, Dutta et al., 2015 found that DO concentrations dropped exponentially in response to each flooding event. They also showed that oxygen partial pressure recovered to its initial value upon the start of the DP. This observation, however, may not hold true for greater depths (Dutta et al., 2015). In a field scale study, Miller et al. (2006) described oxygen and nitrogen species concentrations in a $\sim 2.7$ m sandy loam soil profile during cycles of 4 days of wetting and 4 days of drying. They found that deeper than 0.6 meters below ground surface, aeration was limited compared to the upper parts of the profile and that deeper than 1.5 meters, the vadose zone was mostly anoxic throughout the wetting and drying cycles (Miller et al., 2006).

In this study we examine the effect of hydraulic operation (i.e. wetting-drying periods) on a 6-meter soil profile's biogeo-chemical dynamics through a series of long-column experiments. We hypothesize that in vadose zones deeper than $\sim 1$-1.5 meters, oxidation-reduction potential (ORP) and DO dynamics differ greatly compared to the shallow parts of a soil profile and are affected to a different extent by changes in hydraulic management and inflow composition. We further hypothesize that insufficient DP will have a detrimental effect on the deeper parts of the soil profile and they will eventually result in DO depletion, negative ORP and impaired outflow quality. Correspondingly, increased DP will be especially significant for the deeper parts of the profile which may lead to extension of the aerobic zone to greater depths.

## 2 Materials and Methods

A 6-meter stainless steel column was designed (Fig.1). The column consists of six one-meter modules, each module is equipped with ports for sensors and sampling equipment (see Table 1). The column was packed with soil from the SHAFDAN site according to the soil horizons at the site. The different layers' grain size distribution as well as initial total organic carbon (TOC) content were determined and are described in the supplementary material (Fahl et al., 2014).





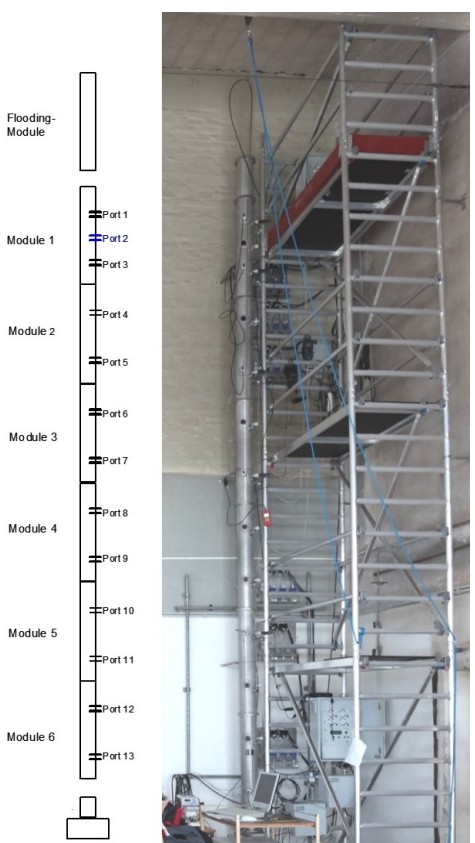

**Figure 1.** The six-meter long column, Dresden, Germany

Prior to the four main column experiments (Table 2), a preliminary flow experiment was conducted in order to roughly estimate the average flow rate through the column as well as the ponding rate. The four main experiments were designed to examine the effect of shorter and longer DP as well as inflow composition on the bio-geo-chemical state and dynamics of the soil-water system at different depths and the ultimate outflow chemical composition (especially DOC and nitrogen species).

The first experiment involved a simple inflow solution of only ammonium ($NH_4^+$; added to tap water, final concentration of $\sim 5$ mg/L), where the hydraulic operation consisted of cycles of 60 minutes of flooding followed by 150 minutes of drying. The term 'flooding period' (FP) refers to the duration of time in which water are pumped to the top of the column. A 'drying period' (DP) starts as the pump is turned off and ends at the beginning of a new flooding event. These specific flooding and drying periods were chosen in light of the preliminary experiment's results , in which it was shown that a DP of 150 minutes is

sufficient for the ponding to infiltrate and allows around 50-60 minutes of free aeration (i.e. no ponding). One of the goals of the preliminary as well as the first experiment was to "awaken" the microbial community of the soil system.

The second experiment involved the preparation of a synthetic TWW inflow solution which included ammonium, glucose and asparagine dissolved in tap water (for exact solution composition - see supplementary material). The synthetic TWW composition was designed to include a moderate-to-heavy load of DOC as well as organic and inorganic nitrogen species





around the concentrations found in the SHAFDAN ponds. Untraditionally, Glucose was chosen as the main carbon source to allow the investigation of the system's behavior around the ranges of ORP values that are found in field SAT systems at various depths and under different hydraulic loads and operation regimes (Orgad et al., 2017). Hydraulic operation for the second experiment included two parts - we first applied cycles of 60 minutes of flooding and 150 minutes of drying, and after 9 cycles the DP were increased to 240 minutes.

During the third and fourth experiments, rael TWW water from the Dresden-wastewater treatment plant (WWTP) were used. Hydraulic operation included 60 minutes flooding periods for both experiments and DP were 150 minutes in the third experiment and 240 minutes in the fourth. During all experiment, sensors' information was recorded and pore solution samples were taken from depths of 25, 75 and 175 cm below soil surface as well as from the outflow. Inflow solution was also sampled and tested to confirm that no major changes in its composition occurred during the experiments.

**Table 1.** Inflow composition and flooding / drying periods in the four discussed experiments

|  | Experiment 1 | Experiment 2 | | Experiment 3 | Experiment 4 |
|---|---|---|---|---|---|
| Inflow | Ammonium solution | Synthetic WW* | | Real WW* | Real TWW* |
| FP / DP (min) | 60/150 | part 1 | 60/150 | 60/150 | 60/240 |
|  |  | part 2 | 60/240 |  |  |

*  Chemical composition is in the supplementary material

For the synthetic TWW solution the following chemicals were used : L-Asparagine anhydrous ($H_4H_8N_2O_3$, $> 99.5\%$, Sigma Aldrich), Ammonium chloride $NH_4Cl$ (Jenapharm-Laborchemie APOLDA, Analysis pure), D(+)-Glucose monohydrate $C_6H_{12}O_6 \bullet H_2O$ (VWR chemicals PROLABO). Exact composition is described in the supplementary material.

Solutions were prepared by slowly adding the appropriate amount of powdered chemical into the needed amount of distilled water (DW) while continuously stirring the solution using a magnetic stirrer. Large amounts of synthetic TWW ($\sim$200 liter for

experiments 1 and 2) were prepared by adding concentrated solutions of the desired chemicals into the remaining amount of tap water (200 liter minus the amount added as concentrated solutions) so that the final desired concentrations were reached. The synthetic TWW solution was then gently mixed. The real TWW used for experiments 3 and 4 was collected from the Dresden (WWTP). After initial chemical analysis, glucose and ammonium were added to match the ammonium, TKN and DOC concentrations to these of the synthetic TWW (see supplementary material).

Sensors along the column included frequency domain refractometers (FDR) sensors for soil moisture measurement were SM300 (Delta-T Devices Ltd ), Tensio 150 (UGT GmbH) tensiometers for presure head, submersible Level Sensor (Sensortechnics) was used for Surface head. ECO Tech Bonn (1.5cm diameter) ceramics were used as suction cups for sampling, LDO10101 (Hach-Lange, Germany) were used for luminescence dissolved oxygen (LDO), and Harburg (ELANA Boden Wasser Monitoring) sensors were used to monitor ORP.



**Table 2.** Sensors' position along the column

| Soil surface | SH sensor | - | - | - |
|---|---|---|---|---|
| 25 cm | FDR | LDO | Suction cup | ORP |
| 75 cm | FDR | LDO | Suction cup | ORP |
| 175 cm | FDR | LDO | Suction cup | |
| 275 cm | FDR | LDO | - | - |
| 375 cm | FDR | LDO | - | - |
| 575 cm | FDR | LDO | - | - |

To asses the chemical composition of the inflow solution as well as the collected samples, four types of chemical analysis were performed: ammonium was measured using ammonium test kit and a Nova 30 Spectroquant (Indophenol blue method). $NO_2$ and $NO_3^-$ concentrations were measured using an Ion chromatograph (IC) after samples were passed through a 0.2 $\mu m$ filter. Total Kjeldahl Nitrogen (TKN) was determined using the standard selenium method (ISO 5663:1984). Dissolved organic carbon (DOC) was determined using the standard method (ISO 8245:1999) after samples were passed through a 0.45 $\mu m$ filter

and HCl was added to prevent any further organic matter consumption by bacteria.

## 3   Results and discussion

Figure 2 presents the WC and DO concentrations at four different depths along the profile (25, 75, 175, 275 cm below soil surface), during 8 flooding and drying cycles of 60 minutes of flooding and 150 minutes of drying (experiment 2, part 1; WC data in a similar manner to Fig.2 for the part 2 of experiment 2 is presented in the supplementary material). The end of each

60-minutes long flooding period is indicated by the 8 peaks of the surface head in each cycle (Fig.2a). It is important to note that the soil surface is covered by water for longer than the 60 minutes of flooding - roughly 130 minutes in each cycle, which means the time where the soil surface is exposed to the atmosphere is roughly 80 minutes. In the second part of this experiment, where DP were 240 minutes, soil surface was actually exposed to the atmosphere for ∼150 minutes during each cycle (see supplementary material). Water front progression through the profile can be retraced by timing the first WC increase following

the first FP. At a depth of 25cm, WC initial increase occurred less than a minute after the beginning of the first FP, while at depths of 75, 175 and 275 cm, it occurred after ∼22,∼53 and ∼90 minutes, respectively. This indicates that flow rate decreased during the first few minutes of the experiment (as should be expected following classic infiltration theory) but stayed relatively constant as the experiment progressed. WC patterns were significantly different between the various depths - while at a depth of 25 cm WC values ranged between 20.5% and 31.2%, at a depth of 275 cm below the surface, the maximal WC was 18.4% and

it dropped below 12% following each DP. Haaken et al. (2016) followed WC patterns in one of the SHAFDAN's infiltration ponds using electrical imaging. While their work is in the field scale, differences in WC between different depths of the profile agree with our findings (Haaken et al., 2016).





DO concentrations also displayed different patterns at different depths of the profile (Fig.2). In the upper parts of the profile (25cm depth), DO concentration increased in response to each of the DP. In each cycle, the drop in WC caused by the beginning

of the DP led to recovery of the DO concentrations and ultimately to complete DO saturation (Fig.2b). As observed in multiple studies in laboratory and field work, close to the surface, DO concentrations are expected to increase in response to the soil aeration during the DP (Mienis et al., 2018; Miller et al., 2006). Regardless of the oxygen movement mechanism (diffusion, advection or convection), the short distance assures fast response of the system.

At 75 cm depth (Fig.2c), while DO recovery is still observed in response to the beginning of DP, aeration is much less

effective and DO concentration averages (through the 8 cycles) dropped by $\sim 2.68$ mg/L. During the two first cycles, DO concentrations remained almost constant and close to saturation, but upon the beginning of the third FP, a significant decrease was observed. This decrease resulted in a 25% drop in DO concentration following all the next DP. After this initial drop, however, DO patterns and amplitude (following each DP) were sustained through the remaining cycles.

In the deeper parts of the column (Figs.2d and 2e), DP induced DO recovery to a much smaller extent. DO concentrations

were able to remain relatively high and constant through the first three cycles, likely due to lower nutrient consumption rates by bacteria at these depths, compared to the upper parts of the profile (Quanrud et al., 1996). Note that DO drop at those depths is a bit hindered, compared to the upper layers. However, as the experiment progressed, DO concentrations decreased steadily to almost complete depletion. These results are to be expected in light of previous laboratory and field observations that demonstrated a decline in DO concentrations with depth (Miller et al., 2006; Orgad et al., 2017). Moreover, DO concentrations along

a SAT system's profile are affected by both air movement patterns through the profile during the DP and the consumption of oxygen by bacteria. Since bacterial activity was previously shown to remain relatively unchanged during desiccation(Roberson et al., 1992; Zhang et al., 2012), DO recovery is highly dependent on oxygen transport mechanisms during the DP. These advective and diffusive processes exposure of the surface to the atmosphere, but it also depends on path that air has to pass through, which is longer for greater depths.





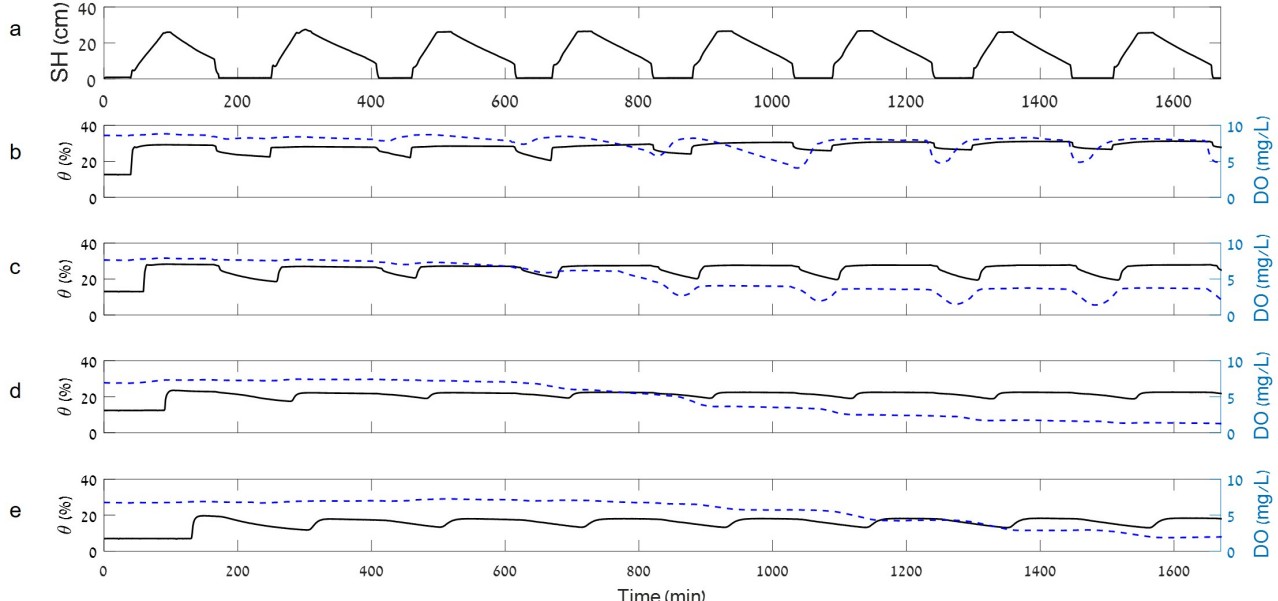

**Figure 2.** Surface head (SH), water content ($\theta$) and DO over time at depths of : 25, 75, 175 and 275 cm below soil surface during part 1 of experiment 2 (150 minutes drying)

As reflected from the results of the first part of experiment 2 (Fig.2), the 150-minute DP resulted in significant DO depletion in depths greater than 75 cm. To examine the effect of a change in hydraulic operation (longer DP), we increased the length of the DP to 240 minutes (60% increase) from the 10th cycle. Figure 3 presents the DO concentrations at four different depths in the soil profile through the two-stage flooding-drying campaign (experiment 2 parts 1 and 2) Increasing the DP, we increased the time of soil surface's exposure to the atmosphere to $\sim$ 150 minutes on average (compared to $\sim$ 50 minutes on average for part 1) and thus expected to observe increased aeration and some recovery of DO concentrations deeper than 75 cm (where the shorter DP led to almost full DO depletion). During this part of experiment 2, WC values at a depth of 75 cm below the surface ranged between $\sim$27% and $\sim$17% (compared to $\sim$18% and $\sim$12% for part 1). At depths of 175 and 275 cm below ground surface, maximal WC were $\sim$ 22 and 18 %, respectively, and minimal WC were $\sim$ 16 and 11 % respectively. At 75 cm depth, shortly following the first longer DP ($\sim$ 100 minutes), DO concentrations increased by $\sim$60% and reached an average of $\sim$ 6.4 mg/L following each of the next DP (compared to an average of $\sim$ 3.7 mg/L during the shorter DP, after the initial drop around the third cycle). In the deeper parts of the profile, a delayed, moderate yet significant response to the increase in DP was observed. The delay in response time compared to the DO recovery in the upper most part of the profile, increased with depth, corresponding to the expected dynamics of air and oxygen movement through the soil profile (DO recovery was observed after around 2,700 minutes for 375 cm and 3,700 minutes for 575 cm). The minimum DO concentrations recorded through the two-stage experiment averaged 0.79, 1.32 and 1.62 mg/L for depths of 175, 375 and 575 cm, respectively . These ranges of DO concentrations below 1.5 meters underground are found in many SAT sites around the world (Amy et al., 2007)



and explain the negligible aerobic bacteria activity found under these conditions.

Following each of the longer DP, DO concentrations in the 175 cm sensor fluctuated periodically in response to the wetting
and drying periods and reached 4.1 mg/L following the drying events. In the two deepest sensors, DO concentrations stabilized
around ∼ 3.9 mg/L in the 375 cm sensor and ∼ 3.25 mg/L in the 575 cm sensor. Considering the fact that sustaining the shorter
DP of stage 1 (of experiment 2) would result in total DO depletion ∼ 175 cm depth (supplementary material), these are very
important observations. Clearly, too short DP will lead to reducing conditions, at some depth, which as a consequence may
lead to insufficient degradation of residual DOC, nitrogen species, and possible leaching of undesired minerals and compounds,
such as manganese to the deep vadose zone and the aquifer (Goren et al., 2012) . In addition, the prominent increase in DO in
the deeper parts of the profile indicate that longer DP means not only enhanced oxidizing conditions at a given depth, but also
increase of the profile volume that did not develop anoxic or anaerobic conditions over the flooding and drying cycles.

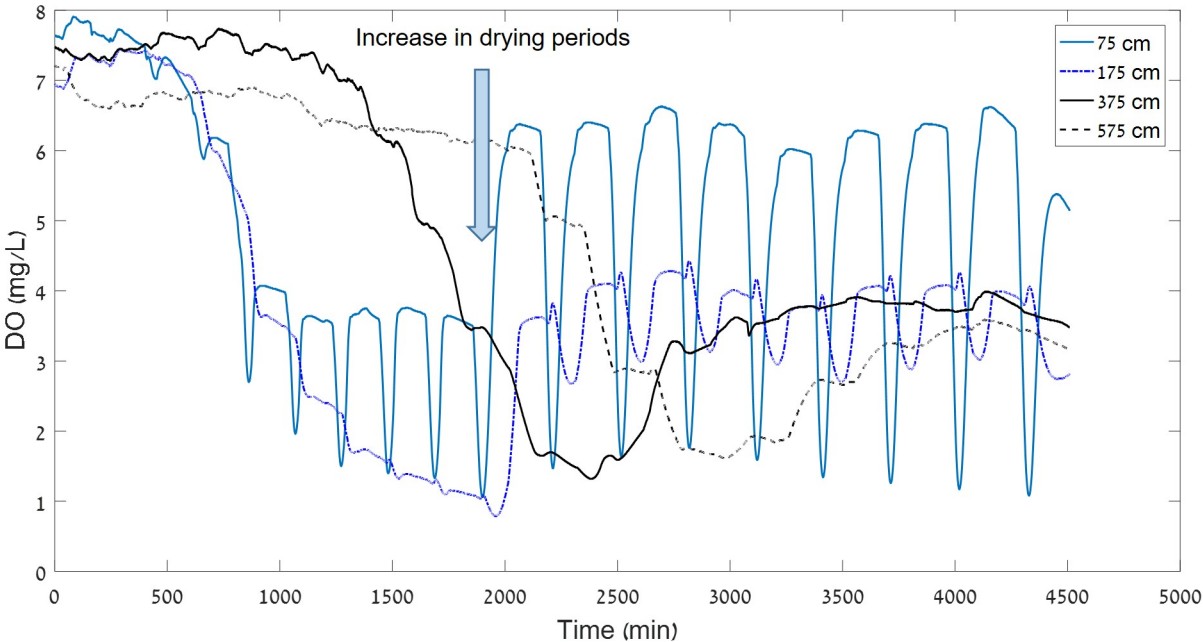

**Figure 3.** DO concentrations during the two stages of experiment 2

Experiments 3 and 4 were designed to further examine the effect of the difference in DP length on the bio-geo-chemical
state of the profile and its effect on some water quality parameters. In these experiments we used real TWW, collected from
the Dresden wastewater treatment plant (WWTP) after the activated sludge process. Similarly to the observed in experiment
2, longer drying periods had an advantage in terms of DO concentrations along the profile (not shown). ORP measurements
in the upper parts of the column (75 cm) revealed that ORP was significantly higher during experiment 4, ranging between ∼
+400 and ∼ +160 mV compared to experiment 3, during which ORP was mostly negative and reached values as low as ∼ -530





mV (supplementary material).


Figure 4 presents DOC, TKN and ammonium concentrations at the outflow (i.e. 6 meters below soil surface), comparing the results for experiments 3 (DP of 150 min) and 4 (DP of 240 min). Outflow ammonium, DOC and TKN concentrations during experiment 4 were significantly lower compared to their inflow concentrations ($\sim$ 0.04, $\sim$ 1.65 and $\sim$ 0.62 mg/L respectively). During experiment 3, ammonium, DOC and TKN outflow concentrations were also lower compared to the inflow, but averaged significantly higher compared to experiment 4 ($\sim$ 4.4 and $\sim$ 3.8 mg/L, respectively; t-test, $\alpha$=0.05), suggesting that the longer DP had a significant positive effect on the outflow quality.

During experiment 3, ammonium concentrations were $\sim$ 0.12 mg/L after $\sim$ 400 minutes from the beginning of the experiment but increased noticeably in the following samples (taken after $\sim$ 1500 minutes) while $NO_3^-$ concentrations decreased correspondingly. Considering the DO concentrations recorded during the experiment, this observation is to be expected. Progressively decreasing DO levels were observed at all depths greater than 75 cm starting after $\sim$ 850 minutes from the beginning of the experiment (see DO data for depths of 175 and 275 cm in the supplementary material). This suggest that the DO depletion caused by restricted aeration for this duration of time, led to decreased rates of nitrogen species' oxidation, which may explains the increase in ammonium concentrations at the outflow after $\sim$1500 minutes. However, looking at the TKN analysis for this experiment, we found that TKN did not significantly increase, suggesting that the increase in ammonium concentrations occurred simultaneously to a decrease in organic nitrogen concentrations. This behavior may theoretically be attributed to increased rates of ammonification. However, ammonifying bacteria populations have been shown to thrive under DO saturation (Ruan et al., 2009), which makes this possibility unlikely. Since this study did not include microbial identification, further investigation is needed in order to reveal the exact nature of this observation.





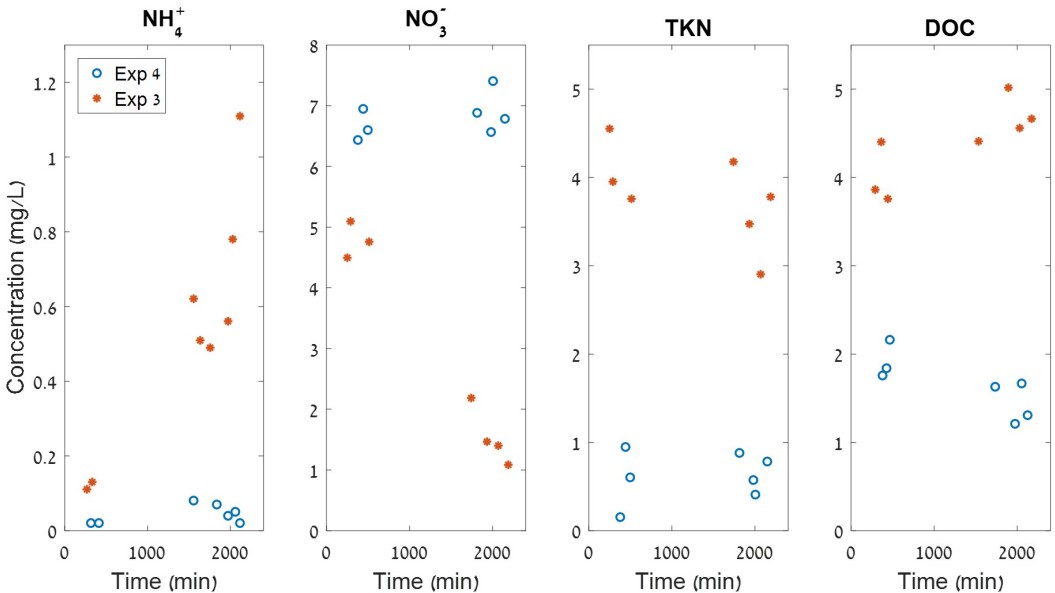

**Figure 4.** Outflow concentrations of ammonium, $NO_3^-$, TKN and DOC (mg/L) , during experiment 3 (red) and 4 (blue)

## 3.1 Comparison with field observations

In a field study conducted in the Israeli SHAFDAN site in 2015-2016, Orgad et al. (2017) recorded ORP values along the wetting and drying cycles in a series of flooding and drying campaigns over a year (Orgad et al., 2017). Figure 5a presents the maximal ORP values in these campaigns at a depth of 95 cm below the pond's surface, around 20 m south-east of the inlet. The ensemble of wetting-drying cycles over an entire year assures a wide variety of wetting and drying proportions. The recorded data showed that when DP were ~15 hours or longer, the maximal ORP values consistently indicated aerobic conditions (and reached ~ + 650 mV).

Figure 5b presents the same quantity (maximal ORP as a function of DP) for the column experiments presented above. Despite the major differences in time scales (order of 2-4 hours for column experiments vs. order of 15 hours for field conditions), TWW composition as well as system's structure and characteristics between our 1D system and a 3D full scale SAT site, our measurements (WC, DO, ORP) qualitatively agree with the field observations. That is, longer DP led to higher (oxidizing) ORP values, while shorter DP led to lower ORP values (reducing conditions). Comparing the ORP values between the field and the column results (Figs.5a and 5b), it is clear that there is a non-negligible difference - while ORP values never exceeded +440 mV in the column experiment, values of ~ +700 mV were observed in the field. Moreover, minimal ORP values did not drop below ~ +100 mV in the field while in the column, negative values of ~ -150 mV were observed. These differences probably stem from the fact that in the column, a 1D system, air (and water) flow only along the vertical direction while in the field, air flows through the soil profile in all directions and thus allow overall enhanced aeration. Further, in the field the





infiltration pond is filled from a single inlet, which means it takes about 10 hours for full surface coverage of the pond. This suggests that further from the inlet, lateral air-flow is possible even hours after the initial flooding.

The difference in time scales between the column and the field experiments may also suggest a difference in the proportions
240 of the air supply mechanisms to different layers of the subsurface. Mizrachi et al. (2016) have suggested that air (in the gas phase) may be "pushed" downwards once the soil surface is fully flooded by water (Mizrahi et al., 2016). While this phenomenon is likely maximized in a column experiment (as the soil-air has no time and no horizontal pathway to be released), this mechanism and its impact were not investigated here. However, it is likely that in field conditions, where lateral air movement is allowed, this mechanism of air supply is of lesser significance, and air movement due to diffusion or air-pressure
245 gradients becomes more dominant.

Interestingly, it is evident from the field data (Fig.5a) that very long DP (>20 hours) did not present an advantage over DP of moderate length (between 15 and 20 hours), at least at a depth of 95 cm. Considering the common understanding that most of the biochemical degradation of organic matter at the SHAFDAN SAT site happens at the upper meter or so (Quanrud et al., 1996), DP of more than 15-20 hours seem to have no influence on the oxidation state of the profile. In light of the understanding
250 that longer DP lead not only to increased aeration but also to an increase of the aerobic volume of the soil profile,

a combination of longer FP and sufficiently long DP (such that allow this profile aerobic volume increase) may be beneficial for both amounts of TWW that the site is able to treat per unit time as well as reclaimed water quality.

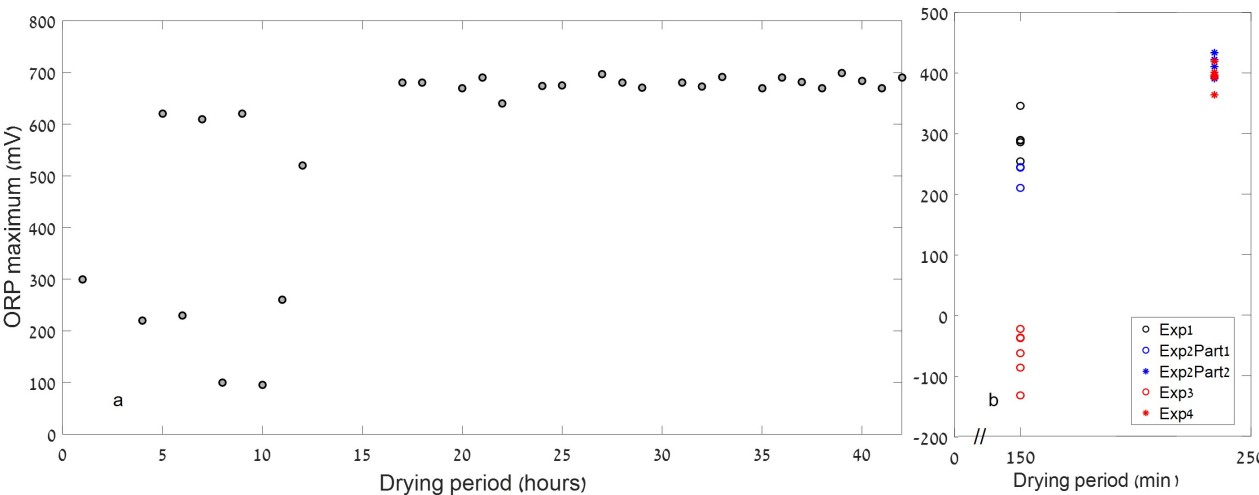

**Figure 5.** Maximal OPR values versus the length of DP. a) during field infiltration campaigns in the SHAFDAN site and b) during the column experiments 1, 2, 3 and 4

As described in Table 1, two of the four experiments presented in this work (experiments 3 and 4) included real TWW while in the two earlier experiments synthetic TWW were used. To allow comparison between the experiments, their DOC, TKN
255 and ammonium concentrations were matched (see supplementary material) so that the main difference between the two WW





sources stems almost entirely from their microbial content (likely there are some differences in micro-nutrients as well). Compared to tap water-based inflow that contained very low concentrations of bacteria, the real TWW collected from the Dresden WWTP had gone through an activated sludge reactor but were not disinfected. Hence, it contained much larger quantities of microorganisms. During experiment 3 (150 minutes of drying, real TWW), we observed lower ORP values compared to the

first part of experiment 2 that is identical in terms of FP/DP. The average peak ORP throughout the first part of experiment 2 were ~+220 mV while in experiment 3, the average peak ORP was ~-80 mV (Fig.5b), despite the fact that the inflow in both experiments contained similar organic loads. From these results, it is clear that the nature of the secondary treatment, which determines the inflow composition (in terms of inflow bacteria concentrations as well as organic load) affects the biogeochemical state of the SAT system through reactions' rates and scope. Our observations suggests that the contribution of the microbial

community in the influent itself is more dominant in the degradation process than commonly perceived. In their review, Sharma et al. (2017) found that recovered TWW's quality in SAT sites around the world varied significantly depending on the inflow source. Removal of DOC, nitrogen species and organic micropollutants was generally better when the pre-treatment was more extensive. Specifically, they found that advanced processes like ozonation, implemented prior to SAT resulted in higher removal efficiency (Sharma et al., 2017). Since ozonation would generally damage the microbial community in the influent, it seems

like the absence of microorganisms in the influent enhances organic matter degradation by soil's native bacteria. However, advanced oxidation processes affect more than the microbial content of the TWW alone and thus, to understand the role of influent microbial community on the biogeochemisrty in SAT systems, further investigation is needed.

**4 Summary and conclusions**

In a series of long column experiments we examined the expected SAT system response to different hydraulic operation regimes, namely wetting and drying periods. Experiments were conducted using real and simulated treated wastewater. WC, DO and ORP were tracked along the column, and water quality samples were collected along the column and at its bottom end, at 6 m, and analysed for nitrogen species and DOC. Hydraulic regimes considered were of 60 minutes of wetting, and

150 or 240 m of drying (corresponding to ~160/50 and ~150/150 minutes of the soil surface being ponded/exposed to the atmosphere, respectively).

As could be expected, shorter drainage periods lead to higher WC regimes in the subsurface, and lower DO values. Interestingly, the shorter DP also lead to almost complete depletion of the DO throughout the column (below ~50 cm), and creation of prevailing low ORP values, indication anoxic to reducing conditions throughout most of the column. Longer DP, on the

other hand, lead to oxidizing conditions throughout the column for most of the time. These conditions are also expressed as column outlet water qualities, where longer DP lead to better oxidation of ammonium, and reduction of total nitrogen as well as DOC. The almost immediate response of the DO concentrations at depth to the surface exposure time to the atmosphere suggests domination of advective oxygen movement in the gaseous phase (over diffusion in the gaseous phase or advection in





the liquid phase). These results suggest that longer DP not only shift the active (oxidizing) part of the system towards oxidizing
conditions, but it also makes this region larger.

Longer DP have their consequences - on one hand, they lead to more oxidizing conditions, and to extension of the oxidizing parts of the variably saturated zone. On the other hand, they lead to reduction in the volumes that can be infiltrated and recharged to the aquifer. Clearly these are contrasting objectives that need to be further explored and optimized for a given site. The qualitative similarity between ORP values in these column experiments and those measured at the SHAFDAN site suggest
that these findings are also relevant for real conditions, and that such optimization can be conducted also in reality.

To conclude, our results suggest that the variably saturated zone in a SAT system can and should be seen as a pseudo reactor, in which DO and ORP values can, to a certain extent, be controlled. Longer DP dictate better degradation of ammonium, organic nitrogen and organic carbon, but lead to reduced infiltration. The immediacy of the DO recovery at depth following sufficient DP may suggest that the ratio between drying and wetting periods is not the parameter that dictates the SAT biogeochemical
dynamics, and long enough DP (that is primarily a function of hydraulic properties and desired oxidizing depth) can be followed by much longer wetting periods. This however was not examined, yet.

*Author contributions.*   FB and JS prepared the experimental set up, SBM, AF and NW designed the experiments, OO provided and analysed the field data, SBM preformed the column experiments, analysed the data and prepared the paper with contribution of all authors.

*Competing interests.*   The authors declare that they have no conflict of interests

*Acknowledgements.*   This work is financed within the framework of the German - Israeli Water Technology Cooperation Program under project number WT1601/2689, by the German Federal Ministry of Education and Research (BMBF) and the Israeli Ministry of Science, Technology and Space (MOST). We wish to thank the German-Israeli Cooperation in Water Technology Research for funding the productive stay in Germany through the Young Scientists Exchange Program (YSEP).



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
