# Peer review of "On the role of operational dynamics in biogeochemical efficiency of a soil aquifer treatment system"

_Hydrology and Earth System Sciences, 2019_

## Referee Comment (RC1) · Anonymous Referee #1 · 6 Oct 2019

6-10-2019

A review of: On the role of operational dynamics in biogeochemical efficiency of a soil aquifer treatment system, by Ben Moshe et al.

Summary and Recommendation

The manuscript describes a series of experiments in which sequences of flooding with treated wastewater (TW) and drying periods (DP) are imposed on the surface of a 6 m deep column, as a physical analog of advanced soil aquifer treatment (SAT) in a naturally- thick unsaturated zone setting (e.g. Shafdan, Israel). The impact of DP on water content (WC) dissolved Oxygen (DO) and oxidation-reduction potential (ORP)

along the column and dissolved organic carbon (DOC) and nitrogen (N) species at the outlet were examined. The important and new result (to the best of my understanding) is that longer DP not only improved the aeration and reactivity in the top 1m which is well known, it can also keep deeper parts of the column (1.75-5.75 m) in aerobic condition (DO = 3 - 4 mg/l) hence more reactive. Some similarity between ORP trends with DP between the column experiment and field experiment in an operational SAT facility are shown.

As managed aquifer recharge (MAR) is growing fast worldwide, and more MAR operations are used also as SAT operational aspects of this cheap water treatments offered by mother earth are of great interest for HESS readership. The finding of possible deep aerobic conditions has significant implications on loads of organics and N that can be treated by SAT. Therefore the manuscript is worthy for publication. Nevertheless the current presentation is far from HESS standards and the paper is hard to read even though there in nothing sophisticated in it. Therefore I recommend major revisions following the comments herein.

Major Comments

1) Emphasize the important result: long DP -> deep aerated reactive interval, throughout the results and discussion (is it first result of its kind?).

2) The absence of reference to the flow in the column is annoying (e.g. flow rates, hydraulic properties of sediments; a simple 1D water flow model; more sophisticated flow of water and air model . . .). It is a controlled experiment in a column filed with porous medium, the hydrologist reader deserves a better acquaintance with this simple flowing system. The times of flooding and drying periods are meaningless without knowing the range of flow rates in the column. A calibrated model and simulations of different DP are a natural continuation of the research starting with the experiment, and can be in a following paper, but no reference of the flow condition in the column is not acceptable. Ponding and drying in a thick unsaturated-zone infiltration system is

needed not only for the biochemistry, but also to sustain infiltration rates (see Ganot, Y., R. Holtzman, N. Weisbrod, I. Nitzan, Y. Katz, and D. Kurtzman. 2017. Monitoring and modeling infiltration-recharge dynamics of managed aquifer recharge with desalinated seawater, Hydrol. Earth Syst. Sci., 21, 4479-4493).

3) Concentration units and naming chemicals entities – be consistent in naming and with units. Micro-molar than mg/l and in the N species is it as N or for the molecule?.I suggest use mg/l as C for DOC and mg/l as N for all N species thought the manuscript and say it explicitly. NO2- is an anion, "ammonium and NO3-", spell the chemical formula for the ammonium as well.

4) Figure captions are laconic. A figure and its caption should be much more standalone entities. For example: Figure 4 has no meaning for a reader without looking for "Experiment 3" in the text, while a few words can make it meaningful. Go over all captions.

5) Supplement - Sediment characteristics should be in the main text as part of dealing with comment # 2. A table of the chemical characteristics of all the water types should also be in the main text.

6) Scientific-writing editing is needed. In many places a reference is referred to in both the beginning and the end of a sentence, synonyms with no explanation in abstract, typos, consistency (part 1 vs. – stage 1) if possible give meaningful names to the experiments – e.g. DP-240-SW or similar is better than meaningless experimant2/stage 2.

Specific comments

1) Abstract. Some numbers describing the main results in the abstract will help. For example in the deep layers DO stabilized on 1- 2 mg/l in the short DP and 3-4 mg/l for the long DP. Also % of removal of DOC TKN for the different DP.

2) L13 – major comment (MC) 6

3) L18 "pseudo" why pseudo? It's a real reactor.

4) L24 MC 6 typo

5) L38 I would say: ... local stream and the Mediterranean sea

6) L41-42 MC 6

7) L51-52 MC 6

8) L52 – explain TKN = organic + ammonium nitrogen

9) L53 MC 3

10) L81 delete "roughly"

11) "Untraditionally" not clear

12) L100 rael->real

13) L104 Table 1 - MC 5, MC 6

14) L105 "H4H8N2O3" should be I believe C4H8N2O3

15) L114-115 MC 3, MC 5

16) L123 TKN defined before

17) Figure 2 caption: 1) what panel for what depth (a, b,c..)? 2) The initial (residual) WC ($\sim$ 15%) lookss high for the sandy sediments in the column, explain.

18) L173 numbers do not fit the figure (12-18%) and not logical, larger DP-> smaller WC makes more sense.

19) Figure 3 - MC 4 (big time). After making the figure +caption a standalone entity I would consider adding. At the caption: "note the convergence of the deep sensors to < 2 mg/l after the short DP versus convergence too > 3 mg/l in response to the long DP." or similar – MC1
20) L203 "(~0.04. . ..." are these the outflow concentrations? The inflow are orders of magnitude higher. Clarify.

21) L204 missing a concentration (for NH4 I believe)

22) L 220-221 MC 6.

23) L 240-241 MC 6

24) L252 Long FP means infiltration rates will decrease due to wetting front reaching some less permeable layers at depth. Draining the top sandy layers is essential also for maintaining high infiltration rates not only for the biochemistry.

25) Figure 5 – in what depth is the ORP probe at Shafdan? MC 4

26) L278 delete "quality"

27) L296 Why "pseudo"? same as comment #3.

Please also note the supplement to this comment:
https://www.hydrol-earth-syst-sci-discuss.net/hess-2019-371/hess-2019-371-RC1-supplement.pdf

---

## Referee Comment (RC2) · Anonymous Referee #2 · 10 Oct 2019

1) General comments

The study carried out by the authors is of deep interest in the context of water scarcity and re-use in arid and semi-arid regions of the globe. The increased use of soil aquifer treatment (SAT) definitely urges soil and water scientists to acquire a better understanding of such complex systems. The content of the presented manuscript is therefore worthy of publication in a journal such as HESS. Studying SAT is not trivial as it deals both with soil hydrology and the science of water treatment. The experts in the aforementionned fields will however have an hard time reading this manucript. As an hydrologist, one will find himself frustated because of a poor description of the flow

conditions in this simple 1D-column. The basic notions of soil hydrodynamics are over-looked. Experimental variables such as hydraulic loading rate and saturated hydraulic conductivity of the soil are not mentionned which makes any comparisons with other studies complicated and makes it hard for the reader to understand initial and boundary conditions. In addition, the use of vague terms and notions such as flow rates, timing water content (WC) peaks or time to replenish oxygen concentration (instead of expressing mean water velocity or reoxygenation rate) is not acceptable. As an expert in water treatment technologies, one will find himself exasperated by the absence of a proper description of the biogeochemical parameters (e.g. characteristics of the wastewater such as chemical and biochemical oxygen demand, total suspended solids per liter of water, number of colony forming units per liter of water....) and by the improper use of units (see specific comments section). Such information should be mentionned and properly summarised in the main body of the article (not in the supplementary material) considering that they are the most important experimental variables affecting the results. The experimental design of this study is quite impressive and definitely attracted my attention. However, it is disappointing that the take-home message of the study is quite trivial (i.e. longer drying periods allow for higher ORP values but mean less volume of water infiltrated per unit of time). The other conclusions are somehow weak and not put in a straigthforward manner. In addition, the train of thoughts of the authors is most of the time unstructured which makes this manuscript hard to read. The efforts made to carry out this study definitely should result in a greater contribution to the topic of management and operation of SAT. Considering the above and the comments listed below in the appropriate sections, I recommend major revisions of this manuscript.

2) Specific comments

1. (line 95) - What is the link between choosing glucose as the main source of carbon and the fact that enables the study of the behaviour of the system in field SAT ? Why is it not tradional ? Information is missing or this sentence should be restructured.

1. (line 102) - What was the frequency of data acquisition by the sensors ? As a subsequent question, was there any data manipulation/processing (e.g. outlier removal, filtering and/or curve smoothening techniques) of the time series presented in the paper ? If yes, they should be described or at least mentionned. I am really impressed by the quality of the data. At first glance, the time series looked like modelling results to me.

3. (line 115 to 119) - The authors mention the presence of pressure head sensors and soil solution sampling devices. Yet, no data regarding those sensors are shown. Why ? If the authors do not intend to show results, there is no need to mention their presence in my opinion unless it impacted the obtained results (e.g. disturbance of the flow regime at specific location, air intrusion,...).

4. (table 2) - Many space wasted and not many information contained in this table. If a proper (and scaled) schematic of the column was presented in figure 1, this table could be discarded.

4. (line 126) - Comments valid for the whole "Results and discussion part". Since ORP values and oxygen transfer are investigated, it would make sense in my opinion to express WC in terms of relative saturation of water (WC divided by WC at saturation). By doing that, the reader can directly have an idea of which fraction of the pore space is either air-filled or water-filled. Same can be said regarding oxygen concentration which could be expressed as DO/DOsat if the temperature is known at any time of measurement.

6. (line 203) - The following holds true for the entire manuscript. The authors should pay extra attention to the use of units, specifically the ones for nitrogen species. What is expressed here ? milligrams of ammonium per liter of water OR milligrams of nitrogen in the form of ammonium per liter of water ? I suspect the latter but this should be clearly stated (especially in figure 4 where having a common y-axis for all subfigures is simply wrong!). If it is the latter, the notation should be NH4-N (mgN/l) for ammonium

and NO3-N (mgN/l) for nitrate.

1. (line 220) (3.1 Comparison with field observations). The Israeli SHAFDAN SAT site is very poorly (if at all) described in the method section which makes comparisons difficult to interpret. Where is it exactly ? What is the mean annual temperature there ? Under which conditions is it operated ? How is it comparable to the lab experiment conducted in Saxony ? If the point is to make a reliable comparison between the lab and field experiments, extra information should be added and this should be stated clearly as one of the purpose of the study in the introduction part.

3) Technical comments

see in text of the attached pdf file.

Please also note the supplement to this comment:
https://www.hydrol-earth-syst-sci-discuss.net/hess-2019-371/hess-2019-371-RC2-supplement.pdf

[Figure]

**Supplement:**

[revised manuscript text omitted]

---

## Author Comment (AC1) · 11 Oct 2019

We would like to thank Anonymous Referee 1 for the comments. We will account for them in a revised version of the paper, as we report in the following point–by–point reply:

**Major comments (MC)**

**MC 1 -***Emphasize the important result: long DP → deep aerated reactive inter-*

*val, throughout the results and discussion (is it first result of its kind?).*

**Autors' response -** To the best of our knowledge, the presented results are the first to specifically show deep aeration following long DP in a SAT system. We, therefor, added this statement in the 'summary and conclusions' section. In addition, these findings are emphasized in the abstract, discussion and conclusions (L15, L190 and L289 respectively).

**MC 2 -***The absence of reference to the flow in the column is annoying (e.g. flow rates, hydraulic properties of sediments; a simple 1D water flow model; more sophisticated flow of water and air model...). It is a controlled experiment in a column filed with porous medium, the hydrologist reader deserves a better acquaintance with this simple flowing system. The times of flooding and drying periods are meaningless without knowing the range of flow rates in the column. A calibrated model and simulations of different DP are a natural continuation of the research starting with the experiment, and can be in a following paper, but no reference of the flow condition in the column is not acceptable. Ponding and drying in a thick unsaturated-zone infiltration system is needed not only for the biochemistry, but also to sustain infiltration rates (see Ganot, Y.,R. Holtzman, N. Weisbrod, I. Nitzan, Y. Katz, and D. Kurtzman. 2017. Monitoring and modeling infiltration-recharge dynamics of managed aquifer recharge with desalinated seawater, Hydrol. Earth Syst. Sci., 21, 4479-4493).*

**Autors' response -** We fully accept the comment regarding the fluxes and added the appropriate values accordingly. Additionally, the manuscript of Ganot et al. (2017) is indeed important, and we have added reference to it in the introduction.
A numerical model,including water flow, solute transport, air movement as well as the main biogeochemical processes involved in the system was developed and calibrated. The results will be discussed in a separate manuscript that will hopefully be completed soon.
**MC 3 -** *Concentration units and naming chemicals entities – be consistent in naming and with units. Micro-molar than mg/l and in the N species is it as N or for the molecule?.I suggest use mg/l as C for DOC and mg/l as N for all N species thought the manuscript and say it explicitly. NO2- is an anion, "ammonium and NO3-", spell the chemical formula for the ammonium as well.*

**Autors' response -**  According to the suggested, we made sure the chemical formula of ammonium is used throughout the text, with a few necessary exceptions in the M&M (i.e. "Ammonium test kit", "Ammonium chloride"). Concentration units of the results are consistently presented in mg/L, however, in the introduction we included some SHAFDAN concentration data in $\mu M$ (the units used in the cited work). Since these numbers include analysis results and not only specific species (for example - DOC), we'd rather avoid the assumptions that are needed for the unit conversion.

**MC 4 -** *Figure captions are laconic. A figure and its caption should be much more standalone entities. For example: Figure 4 has no meaning for a reader without looking for "Experiment 3" in the text, while a few words can make it meaningful. Go over all captions.*

**Autors' response -**
The captions of all figures and tables were revised. The captions of Figures 2, 3, 4 and 5 were improved.

**MC 5 -***Supplement - Sediment characteristics should be in the main text as part of dealing with comment # 2. A table of the chemical characteristics of all the water types should also be in the main text*.

**Autors' response -**

[Figure]

According to Referee 1's suggestion, water analysis results for the synthetic as well as the real TWW were moved from the supplementary material to the main text (Table 2 in the revised text). After careful consideration, we still believe that soil's characterization data belongs in the supplementary material for simplicity .

**MC 6 -** *Scientific-writing editing is needed. In many places a reference is referred to in both the beginning and the end of a sentence, synonyms with no explanation in abstract, typos, consistency (part 1 vs. – stage 1) if possible give meaningful names to the experiments – e.g. DP-240-SW or similar is better than meaningless experimant2/stage 2.*

**Autors' response -**
Scientific writing revision was performed for the manuscript. All the specific comments (SC) regarding writing editing were addressed. Experiments' names were changed to describe the DP and WW used (e.g. experiment 4 that involved DP of 240 min and real WW will be noted as - RW240)

**Specific comments (SC)**

**SC 1 -** *Abstract. Some numbers describing the main results in the abstract will help. For example in the deep layers DO stabilized on 1- 2 mg/l in the short DP and 3-4 mg/l for the long DP. Also % of removal of DOC TKN for the different DP.*

**Autors' response -**The abstract was re-edited. The revised version includes numerical values of the comparison between the DPs in terms of DO as well as water quality parameters.

**SC 2 -** *L13 – major comment (MC) 6*

**Autors' response -** Corrected according to the comment (Addressed in MC 6).

**SC 3 -** *L18 "pseudo" why pseudo? It's a real reactor.*

**Autors' response -** A classic reactor typically is seen as a well-controlled, fully engineered and completely mixed system. We use the term 'pseudo-reactor' here to distinguish SAT from such reactor.

**SC 4 -** *L24 MC 6 typo*

**Autors' response -** Corrected according to the comment (Addressed in MC 6).

**SC 5 -** *L38 I would say: local stream and the Mediterranean sea*

**Autors' response -** We accepted referee's suggestion.

**SC 6 -** *L41-42 MC 6*

**Autors' response -** Corrected according to the comment (Addressed in MC 6).

**SC 7 -** *L51-52 MC 6*

**Autors' response -** Corrected according to the comment (Addressed in MC 6).

**SC 8 -** *L52 – explain TKN = organic + ammonium nitrogen*

**Autors' response -** An explanation for the term TKN was added.

**SC 9 -** *L53 MC 3*

**Autors' response -** Corrected according to the comment (Addressed in MC 6).

**SC 10 -** *L81 delete "roughly"*

**Autors' response -** Corrected according to the comment.

**SC 11 -** *"Untraditionally" not clear*

**Autors' response -** The reasoning behind the use of glucose as the carbon source in the synthetic wastewater is explained in L96-98. However, we accept that the use of the word 'Untraditionally' is not necessary and hence it was omitted.

**SC 12 -** *L100 rael→real*

**Autors' response -** Corrected according to the comment.

**SC 13 -** *L104 Table 1 - MC 5, MC 6*

**Autors' response -** Was addressed in MC 5 and MC 6.

**SC 14 -** *L105 "H4H8N2O3" should be I believe C4H8N2O3*

**Autors' response -** Corrected according to the comment.

**SC 15 -** *L114-115 MC 3, MC 5*

**Autors' response -** Was addressed in MC 3 and MC 5.

**SC 16 -** *L123 TKN defined before*

**Autors' response -** Corrected according to the comment.

**SC 17 -** *Figure 2 caption: 1) what panel for what depth (a, b,c..)? 2) The initial (residual) WC ($\sim$ 15%) looks high for the sandy sediments in the column, explain.*

**Autors' response -** 1) Caption was improved and the letters (a-e) were associated with the corresponding parameters. 2) Albeit the fact that the soil profile is mostly sandy, it has non-negligible silt and clay content (see Table S5). Additionally, since the DP are not long enough for complete drying of the soil profile, the measured data doesn't reflect the residual WC even in the end of the DPs.

**SC 18 -** *L173 numbers do not fit the figure (12-18%) and not logical, larger DP $\rightarrow$ smaller WC makes more sense.*

**Autors' response -** We thank referee 1 for the attention. We corrected the numerical values.

**SC 19 -** *Figure 3 - MC 4 (big time). After making the figure +caption a standalone entity I would consider adding. At the caption: "note the convergence of the deep sensors to $<$ 2 mg/l after the short DP versus convergence too $>$ 3 mg/l in response to the long DP." or similar – MC1*

**Autors' response -** Was addressed in MC 4. We thank Referee 1 for the caption addition suggestion.

**SC 20 -** *L203 "(∼0.04..." are these the outflow concentrations? The inflow are orders of magnitude higher. Clarify.*

**Autors' response -** These are indeed outflow concentrations, it is mentioned in the text (L202). To make it clearer, the sentence was improved: "Outflow $NH_4^+$, DOC and TKN concentrations during experiment RW240 ($\sim$ 0.04, $\sim$ 1.65 and $\sim$ 0.62 mg/L respectively) were significantly lower compared to their inflow concentrations"

**SC 21 -** *L204 missing a concentration (for NH4 I believe)*

**Autors' response -** We thank referee 1 for the attention. Corrected according to the comment.

**SC 22 -** *L 220-221 MC 6.*

**Autors' response -** Corrected according to the comment (Addressed in MC 6).

**SC 23 -** *L 240-241 MC 6*

**Autors' response -** Corrected according to the comment (Addressed in MC 6).

**SC 24 -** *L252 Long FP means infiltration rates will decrease due to wetting front reaching some less permeable layers at depth. Draining the top sandy layers is*

*essential also for maintaining high infiltration rates not only for the biochemistry.*

**Autors' response -** Addressed in MC 2.

**SC 25 -** *Figure 5 – in what depth is the ORP probe at Shafdan? MC 4*

**Autors' response -** The depth of the field ORP measurements is mentioned in L223. However, we added it to the caption of figure 5.

**SC 26 -** *L278 delete "quality"*

**Autors' response -** Corrected according to the comment.

**SC 27 -** *L296 Why "pseudo"? same as comment #3*

**Autors' response -** Addressed in SC 3.

---

## Author Comment (AC2) · 22 Oct 2019

**In light of the comments of Referee #2 regarding unit use, we re-considered comment MC3 by referee #1. Accordingly, we now accept the original comment:**

**MC3:** *Concentration units and naming chemicals entities – be consistent in naming and with units. Micro-molar than mg/l and in the N species is it as N or for the molecule?.I suggest use mg/l as C for DOC and mg/l as N for all N species thought the manuscript and say it explicitly.  $NO2-$ is an anion, "ammonium and $NO3-$", spell the chemical formula for the ammonium as well.*

[Figure]

**Author's response:** According to the suggested, we made sure the chemical formula of ammonium is used throughout the text, with a few necessary exceptions in the M&M (i.e. "Ammonium test kit", "Ammonium chloride"). Concentration units of the field data (SHAFDAN) presented in the introduction were converted from $\mu$M to mg/L. DOC and TKN analyses are reported in our work in mg/L (of C and N respectively). For $NO_3^-$ and $NH_4^+$, we accept that consistent use of units is preferable and hence we now use $NO_3^- - N$ and $NH_4^+ - N$ in mg/L as was suggested.

---

## Referee Comment (RC3) · Anonymous Referee #3 · 23 Oct 2019

This article is about experiments on 4 long-columns allowing determining the consequences of application of flooding and drying periods more or less long on the redox potential, oxygenation dynamics and the quality of the outgoing treated water. A comparison of the experimental values obtained with a Soil Aquifer Treatment (SAT) for oxygen was then performed in order to validate or not the behavior observed in the laboratory.

This article is interesting in that studying this type of system in the context of the reuse of treated wastewater is important in a context of water scarcity and reuse in arid and semi-arid regions of the globe. Nevertheless, there are weaknesses in this manuscript

especially in terms of the description and the precision of the biogeochemical parameters and units. In addition, one of the most important phenomenon, from my point of view, is the issue of SS that is not addressed at all in this article while it is the main problem when applying treated wastewater on a soil (clogging). I recommend major revisions of this manuscript.

General comments

In general English and spelling (words are often singular when they should be plural) should be reviewed for a better reading of the article. Put dots for numbers and not comma (,)

When we talk about dissolved oxygen, it is better to write its unity in mgO2/L instead of mg/L for better understanding.

Generally, when we talk about nitrogen, concentrations are expressed in mgN/L. Is this the case in this article? For example, Figure 4 shows values but the indicated parameters are NH4+ and NO3-. Is it NH4-N and NO3-N?

Specific comments

Introduction

Lines 26-27: the units used for DOC, ammonium and organic nitrogen are not expressed in the system of international units (mg/L)

Lines 40-42: repetition of Goren et al. (2014)

Lines 51-52: repetition of Mienis et al. (2018)

Materials and methods

Line 78: the reference to Table 1 is not good. Table 1 does not refer to sensors and sampling equipment but to the characteristics of the applied water as well as to the duration of the flooding and drying phases.

Table 1 and Table 2 must be reversed.

Lines 95-97: the sentence should be rewritten to be clearer.

Line 100/Table 1: why call the inflow of experiments 3 and 4 "Real TWW" while additions of glucose and ammonium have been made? If the explanation comes later, put it here.

Table 1: in experiment 3, in the line "inflow" it misses the letter "T" because it is treated wastewater that was added and not raw wastewater.

Line 102: the sentence starting with "During all experiment, ..." should be the beginning of a new paragraph because it concerns ALL the expermientations and not only the experiment 3 and 4. Refer to Table 2. By the way, it lacks an S to "experiment".

Lines 112-114: the first sentence has already been mentioned above (line 100) and the second sentence should be after line 100.

Lines 121: remove the ":" which would indicate a list behind whereas here the different compounds and their methods of determination are separated by dots.

Line 121: why do you write "ammonium" and not NH4+ whereas it has already been defined line 85? True for the whole document.

Line 122: it misses the sign "-" behind NO2.

Results and discussion

Lines 140-142: repetition of Haaken et al., 2016

Line 169: you say Âń 50 minutes on average for part 1 Âż whereas you said line 132 Âń 80 minutes Âż. Be consistent.

Line 179: 3 digits after the decimal point for the minutes are not necessary (2.7 minutes instead of 2.700 minutes).

Line 186: Âń around Âż is not necessary because you write "∼ ". Moreover, write "for

the 375 cm sensor" and "for the 575 cm sensor" instead of "in the 375 cm sensor" and "in the 575 cm sensor". Lines 194-195: again, this information has already be written line 100.

Figure 4: it would be better to display the input concentrations on the graphs to better see the differences between input and output for experiments 3 and 4.

Lines 203 and 205: the numbers in the parentheses are the differences between the concentrations measured at the input and those measured at the output for the experiments 3 and 4? I think that it is not wise to express the efficient removal in terms of differences in concentrations but you should rather express these removal efficiencies in terms of percentage.

Line 203: you say that your measurements correspond to what is measured in the full scale SAT site but we have no table, figure, or at least a reference on which your statement is based.

Line 253: Table 1 should be Table 2.

Summary and conclusions

Line 280: 150 minutes or 240 minutes (and not only 240 m which means meter).

Please also note the supplement to this comment:
https://www.hydrol-earth-syst-sci-discuss.net/hess-2019-371/hess-2019-371-RC3-supplement.pdf

---

## Author Comment (AC3) · 28 Oct 2019

**We would like to thank Anonymous Referee 2 for his/her constructive comments. Most of the suggestions and comments were accepted and implemented in the revised version of the paper, as we report in the following point–by–point reply.**

**However, before we start we would like to put this research in the right perspective, from our point of view. SAT research combines earth sciences (hydrology, soil physics) with biochemical processes associated with wastewater treatment (i.e. processes like nitrification, de-nitrification, mineralization, etc).**

[Figure]

The terminology used in each of the disciplines may sound lacking to people from the other. Our perspective is closer to earth/geo sciences, looking at SAT processes without comparison to classic wastewater treatment, rather as processes that may be controlled and manipulated by the system's operational dynamics. We believe that some of the comments provided by the reviewer are due to this difference in perspective. But perhaps more importantly, our perspective in this study is to test the ability to conceptually change (and improve) SAT operation. While we do qualitatively compare our results to the SHAFDAN facility in Israel, the specific details of the site are less important than the concept that SAT sites (both in field and laboratory scale) should not be seen as a passive component of the wastewater treatment process but as a 'pseudo reactor' that may (and should) be controlled by hydraulic operation manipulations.

**General comments (GC)**

**GC 1 -** *... The basic notions of soil hydrodynamics are overlooked. Experimental variables such as hydraulic loading rate and saturated hydraulic conductivity of the soil are not mentioned which makes any comparisons with other studies complicated and makes it hard for the reader to understand initial and boundary conditions*
**Autors' response:** We fully agree - hydraulics were so trivial (to us) that we forgot to include it. Average flux (that is of the same order of hydraulic conductivity in our gravity driven system) is now included in the 'Materials and Methods' section.

**GC 2 -** *...In addition, the use of vague terms and notions such as flow rates, timing water content (WC) peaks or time to replenish oxygen concentration (instead of*

*expressing mean water velocity or reoxygenation rate) is not acceptable.*

**Autors' response:** We agree in part with this comment. Where possible, terms were clarified. However, we do not see some of the terms suggested by the reviewer, adopted from the classic environmental engineering terminology, as being proper to SAT. Therefore we choose to keep some terms and avoid using terms that may be misleading (such as 'reoxygenation rate'), as we later elaborate in our response to the Technical comments (specifically -the technical comment referring to line 177).

**GC 3 -** *As an expert in water treatment technologies, one will find himself exasperated by the absence of a proper description of the biogeochemical parameters (e.g. characteristics of the wastewater such as chemical and biochemical oxygen demand, total suspended solids per liter of water, number of colony forming units per liter of water....) and by the improper use of units (see specific comments section). Such information should be mentioned and properly summarised in the main body of the article (not in the supplementary material) ...*

**Autors' response:** We accept that wastewater chemical analysis data should be in the main text rather than the supplementary material. Therefore, we included this information for both the synthetic and real wastewater as Table 2 in the revised manuscript. However, we see this work as a conceptual attempt to discuss SAT operation and its effect on the biogeochemical state of the soil profile. Hence, the very specifics of the wastewater and soil, while important for the sake of completeness, are not the focus of this manuscript and could add unnecessary complexity to this type of paper.

**GC 4 -** *The experimental design of this study is quite impressive and definitely attracted my attention. However, it is disappointing that the take-home message of the*

*study is quite trivial (i.e. longer drying periods allow for higher ORP values but mean less volume of water infiltrated per unit of time).*

**Autors' response:** This comment helped us understand that the main conclusions of this study were not highlighted well enough. It is true that qualitatively increasing DP will result in better oxygenation of the subsurface. However, the classic way SAT is being looked at is of a system where most of the oxidizing conditions (and hence removal of most of the ammonium and organic matter) happen in the very shallow subsurface. What we show here, we believe for the first time, is that longer DP also means extending the volume of the aerated subsurface, or increasing the volume of the 'pseudo-reactor', in our terminology. In other words, we extend the aerobically-active part of the system. We highlighted this conclusion in the revised 'Summary and Conclusions' section. We expect to further support our conclusions in a follow-up paper that includes the development and calibration of a full numerical flow and reactive transport model.

**GC 5 -** *The other conclusions are somehow weak and not put in a straigthforward manner. In addition, the train of thoughts of the authors is most of the time unstructured which makes this manuscript hard to read. The efforts made to carry out this study definitely should result in a greater contribution to the topic of management and operation of SAT...*

**Autors' response:** We thank the reviewer for this comment. The entire manuscript was revised and we believe it reads much better now. Moreover, in addition to the main points described above that are shown here for the first time in the context of SAT, the work described here assisted to develop a numerical model that will help to improve SAT operation under various conditions. Therefore, there will be a significant overall contribution both scientifically and practically (to SAT operation).

**Specific comments (SC)**

**SC 1 -** *(line 95) - What is the link between choosing glucose as the main source of carbon and the fact that enables the study of the behaviour of the system in field SAT ? Why is it not traditional ? Information is missing or this sentence should be restructured*

**Autors' response:** We accept that the word 'Untraditionally' is not clear and even confusing. Hence, we omitted it. Our original intention was to refer to the fact that glucose is usually not the only carbon source in treated wastewater. Nevertheless, since glucose is easily degradable by bacteria (compared to more complex carbohydrates or humic acids that might be present in wastewater) and is common in wastewater treatment and SAT research, its use as the main carbon source allowed us to sustain the short wetting and drying cycles implemented in our experiments and also work in the desired ORP ranges. We included this explanation in the revised manuscript.

**SC 2 -** *(line 102) - What was the frequency of data acquisition by the sensors ? As a subsequent question, was there any data manipulation/processing (e.g. outlier removal, filtering and/or curve smoothing techniques) of the time series presented in the paper ? If yes, they should be described or at least mentioned. I am really impressed by the quality of the data. At first glance, the time series looked like modelling results to me.*

**Autors' response:** Data acquisition was every 1 minute. This information was added to the 'Materials ans Methods' section of the revised manuscript. The raw data was not manipulated or smoothened. The only processing step that was performed is correction of negative values recorded by the surface head sensor - when soil surface was completely dry the sensor would occasionally read small negative values. These values were set to 0. This is now clarified in the 'Materials and Methods' section of the revised manuscript.
**SC 3 -** *(line 115 to 119) - The authors mention the presence of pressure head sensors and soil solution sampling devices. Yet, no data regarding those sensors are shown. Why ? If the authors do not intend to show results, there is no need to mention their presence in my opinion unless it impacted the obtained results (e.g. disturbance of the flow regime at specific location, air intrusion,...).*

**Autors' response:** We fully accept the comment. The tensiometers and suction cups that were mentioned in the text were indeed used for qualitative verification of the flow and transport processes. However, since their results are not presented in this manuscript, we specifically stated it in the revised manuscript: "Tensio 150 (UGT GmbH) tensiometers for pressure head and ECO Tech Bonn (1.5 cm diameter) ceramics were installed along the column as well. While their data is not shown here, it fully supports our presented findings".

**SC 4 -** *(table 2) - Many space wasted and not many information contained in this table. If a proper (and scaled) schematic of the column was presented in figure 1, this table could be discarded.*

**Autors' response:** We accept the comment. In light of the changes we made in SC 3, this table seems to be of minor value to the reader. It was omitted from the revised manuscript.

**SC 5 -** *(line 126) - Comments valid for the whole "Results and discussion part". Since ORP values and oxygen transfer are investigated, it would make sense in my opinion to express WC in terms of relative saturation of water (WC divided by WC at saturation). By doing that, the reader can directly have an idea of which fraction of*

*the pore space is either air-filled or water-filled. Same can be said regarding oxygen concentration which could be expressed as DO/DOsat if the temperature is known at any time of measurement.*

**Autors' response:** This indeed is a point that we had hard time deciding on. On one hand, as the reviewer states, normalized values may be more beneficial as they provide immediate and direct notation of aeration. On the other hand, most readers, so we feel, are more comfortable with actual water content values . Therefore we choose to leave values as are.

**SC 6 -** *(line 203) - The following holds true for the entire manuscript. The authors should pay extra attention to the use of units, specifically the ones for nitrogen species. What is expressed here ? milligrams of ammonium per liter of water OR milligrams of nitrogen in the form of ammonium per liter of water ? I suspect the latter but this should be clearly stated (especially in figure 4 where having a common y-axis for all sub figures is simply wrong!). If it is the latter, the notation should be NH4-N (mgN/l) for ammonium and NO3-N (mgN/l) for nitrate.*

**Autors' response:** DOC and TKN analyses results are reported in our work in mg/L (of C and N respectively). For $NO_3^-$ - and $NH_4^+$ we initially chose to use mg/L units (mg of the species per liter). However, we accept that consistent use of units is preferable and hence we now use $NO_3^- - N$ and $NH_4^+ - N$ in mg/L as was suggested .

**SC 7 -** *(line 220) (3.1 Comparison with field observations). The Israeli SHAFDAN SAT site is very poorly (if at all) described in the method section which makes comparisons difficult to interpret. Where is it exactly ? What is the mean annual temperature there? Under which conditions is it operated ? How is it comparable to the lab experiment*
*conducted in Saxony ? If the point is to make a reliable comparison between the lab and field experiments, extra information should be added and this should be stated clearly as one of the purpose of the study in the introduction part.*

**Autors' response:** The SHAFDAN sites infiltration ponds' operation regime, location and and characteristics were described in multiple publications before. We, therefore, referred to some of them in the introduction and in the 'comparison with field observations' section (e.g. -Icekson et al., 2011, Goren et al.,2014). Section 3.1 of the manuscript shows qualitative agreement between the field and the columns experiments' results. Since the field and laboratory SAT systems are very different in many ways, and especially scale and dimensionality, this agreement is exceptionally interesting and points to the fact that regardless of the major scale differences, some of our findings (i.e. deep aeration and extension of the aerobically-active zone) are relevant to full scale field SAT systems. In that sense, the SHAFDAN site was merely the inspiration to this chapter and not the focus of it. Nevertheless, to allow the reader easy access to the full information, we included a short description of the SHAFDAN site in the beginning of section 3.1. In addition, a comprehensive description was added to the revised 'supplementary material' document.

**Technical comments**

*Referee 2's technical comments are summarized in the following PDF file:*
*https://www.hydrol-earth-syst-sci-discuss.net/hess-2019-371/hess-2019-371-RC2-*
*supplement.pdf*

**Autors' response:** A full revision of the manuscript was performed. Minor comments (e.g. typos, word selection suggestions etc.) were corrected according to referee's suggestions. More general comments are addressed below:

*Figures - Referee suggested multiple adjustments to the figures.*

**Autors' response:** We carefully considered each of the specific comments and we believe that all figures were improved thanks to Referee 2's constructive comments. Specific changes we made according to the comments are hereby reported:

Figure 1: The labels denoting the modules of the column were omitted and the port positions labels were adjusted to a bigger font.

Figure 2: According to Referee 2's suggestion, we added the depths next to each of the a-e sub-plots.

Figure 3: We accepted Referee 2's suggestion to separate the different stages of the experiment by a dashed line and added a clear label denoting 'stage 1' and 'stage 2'. We accept that a presentation of the x-axis in 'days' might be easier to read for long time-series. However, for a system that operates at cycles of hours with no meaning to day/night (sunlight), we feel that this will not help, rather it will make the presentation cumbersome. For example, our FP will be 1/24 days). Therefore, we would rather keep time units in minutes.

Figure 4: The legend of the figure was corrected according to the comment. However, we disagree with the idea of connecting measurements with a straight line. A line connecting two data points implies that a linear trend is assumed. We do not assume that and thus, we believe that singular data pints are more suitable for this figure.

Figure 5: According to Referee 2's suggestion, we changes the y-axes of both sub-plots (Figure 5 a and b) to have the same range of values. The depth of the field measurements was added to the caption of the figure (note that it is also mentioned in Line 223). However, we disagree with the notion of connecting ORP measurements with a straight line. In addition to the above, in the case of the field data, each point represent an independent infiltration campaign. Hence, connecting the dots would not describe accurately the presented data.

*Line 26 - DOC, $NH_4^+$ and organic nitrogen concentrations of secondary effluent at the SHAFDAN site are presented in $\mu M$. Referee suggested to convert to mg/L*

**Autors' response:** We accept Referee's suggestion. Units were converted to mg/L.

*Line 86 - Referee commented that the terms Flooding periods (FP) and Drying periods (DP) were defined before.*
**Autors' response:** The terms FP and DP were indeed defined before. However, this sentence was specifically phrased to clarify authors' interpretation of the terms as it was used throughout the manuscript. Thus, in this case we believe the current wording is appropriate.

*Line 135 - Authors included timing of the water front. Referee commented that this information is not informative*
**Autors' response:** We accept the comment. This line was omitted.

*Line 136 - Authors mentioned 'classic infiltration theory'. Referee suggested to refer to a specific model*
**Autors' response:** We accept the comment. By 'classic infiltration theory' we intended to refer to simple sharp-front models such as the Green and Ampt infiltration model. We added this information in the revised manuscript.

*Line 147 stated "As observed in multiple studies in laboratory and field work, close to the surface, DO concentrations are expected to increase in response to the soil aeration during the DP since regardless of the oxygen movement mechanism (diffusion, advection or convection), the short distance ensures fast response of the*

*system". Referee commented that this is not new information*
**Autors' response:** This is indeed known information that was previously shown by others. We included this line to emphasize the difference between the expected oxygen recovery behaviour in the shallow parts of the profile compares to the deeper parts (that are discussed in the next paragraph)

*Line 162-163 - Referee commented that the sentence is missing the subject and thus is not meaningful*
**Autors' response:** We thank Referee 2 for the attention. The sentence was corrected.

*Line 177 - Referee suggested to calculate re-oxygenation rate instead of the use of the term 'oxygen recovery'.*
**Autors' response:** We thank Referee 2 for the suggestion. We acknowledge that re-oxygenation rates may be valuable information for the understanding of some reactors or filters that are well-mixed or of fixed volume. In this case, however, the increase in DP in response to the longer DP varied between the different depth of the column. For example - while the deepest parts of the column were able to sustain DO concentrations of $\sim$3 mg/L (during the longer DP experiments), the term 're-oxygenation' does not accurately describe the system's behavior. Further, one of our main findings is the relation between DP and the 'oxidizing volume'. After careful consideration, we believe that the use of the term 'DO recovery' is more suitable for the purpose of the sentence.

*Line 186 - Authors stated: "Considering the fact that sustaining the shorter DP of stage 1 (of experiment 2) would result in total DO depletion $\sim$175 cm depth (supplementary*

*material), these are very important observations. Referee commented that the importance of the sentence is not clear to him/her.*

**Autors' response:** This line expresses one of the important points of our work. Studies have shown before that long DP are beneficial for the upper $\sim$ 1 meters of a SAT profile in terms of DO concentrations and oxidation rates. While this is correct, we demonstrated here that deeper areas (in this specific sentence $\sim$175 cm depth) displayed a significant DO increase in response to the longer DPs. This means that longer DPs lead to extension of the aerobic volume of the SAT 'pseudo reactor'. The referral to the fact that sustaining the shorter DPs would lead to complete oxygen depletion in this depth is important for comparison reasons, but we believe that displaying the figure in the main text does not add additional value to the purpose of the claim.

*Line 205 - Authors reported $\alpha$ value for the statistical t-test performed. Referee suggested to display $p_value$ instead.*

**Autors' response:** In the text, we use phrases such as 'significantly higher concentrations' to denote the statistically significant difference in outflow concentrations between experiments 3 and 4. To provide the reader with the information on the significance level we chose for the tests, we report the $\alpha$ value that was the same for all the concentration pairs examined in the t-tests (i.e. DOC, TKN and $NH_4^+$).

*Line 254 - Authors stated that inflow DOC, TKN and $NH_4^+$ content was matched between the synthetic and the real wastewater. Referee pointed this information should be stated in the 'methods' section.*

**Autors' response:** Although the review provided was very detailed, this was probably missed. This information is stated in the 'Materials and Methods' section (Line 113).

[Figure]

*Lines 262-272 - Referee pointed that this paragraph is too vague and hard to follow.*
**Autors' response:** This section was completely revised. The revised paragraph includes a comparison of our findings to a paper by Ak et al.,2013, that compared organic matter removal in a series of column experiments with synthetic and real WW. We discuss the similarities between their results and our findings and also the differences and the possible reasons for them. We believe the revised paragraph is much clearer and better reflects the concept it addresses.

*Summary and conclusions - Referee pointed that there is a change in tense between the first and second paragraphs.*
**Autors' response:** We thank referee 2 for the attention. The 'Summary and Conclusions' section was fully revised and all comments were addressed

---

## Author Comment (AC4) · 4 Nov 2019

We would like to thank anonymous Referee 3 for his/her constructive comments. We will account for them in a revised version of the paper, as we report in the following point–by–point reply:

**General comments (GC)**

**GC 1 -** *One of the most important phenomenon, from my point of view, is the*

*issue of SS that is not addressed at all in this article while it is the main problem when applying treated wastewater on a soil (clogging).*

**Autors' response -** We agree that this is indeed an important topic and its investigation is crucial for SAT sustainability. As reflected by the consistency in surface head and WC patterns along the flooding and drying cycles - we did not observe significant clogging in our system and hence we did not discuss it in the paper. However, since we very much agree that in field scale (or real) systems clogging is a major issue - we now shortly discuss it in the 'Comparison with field observations' section.

**GC 2 -** *In general English and spelling (words are often singular when they should be plural) should be reviewed for a better reading of the article. Put dots for numbers and not comma.*

**Autors' response -** According to Referees' general and specific comments, the entire text was revised. Grammar and spelling mistakes spotted by the Referees or found by the authors in the revision process - were corrected.

**GC 3 -** *When we talk about dissolved oxygen, it is better to write its unity in mgO2/L instead of mg/L for better understanding.*

**Autors' response -** We agree that the presentation of concentrations should indicate the correct species measured by the measuring analytical tool used / sensing device. However, as oxygen is dissolved in water as $O_2$, it is very acceptable and common to present its concentrations as mg/L (given that the species is noted as DO). We agree that Referee's suggestion is also a valid form of presentation but in this case we choose to leave the notations as they are currently presented.

**GC 4 -** *Generally, when we talk about nitrogen, concentrations are expressed in mgN/L. Is this the case in this article? For example, Figure 4 shows values but the indicated parameters are NH4+ and NO3-. Is it NH4-N and NO3-N?*

**Autors' response -** We accept Referee's suggestion and we now use $NO_3^- - N$ and

$NH_4^+ - N$ in mg/L .

**Specific comments (SC)**

**Introduction**

**SC 1 -** *Lines 26-27: the units used for DOC, ammonium and organic nitrogen are not expressed in the system of international units (mg/L)*
**Autors' response -** We accept Referee's comment. Units were converted to mg/L.

**SC 2 -** *Lines 40-42: repetition of Goren et al. (2014)*
**Autors' response -** Corrected according to comment.

**SC 3 -** *Lines 51-52: repetition of Mienis et al. (2018)*
**Autors' response -** Corrected according to comment.

**Materials and Methods**

**SC 4 -** *Line 78: the reference to Table 1 is not good. Table 1 does not refer to sensors and sampling equipment but to the characteristics of the applied water as well as to the duration of the flooding and drying phases.*
**Autors' response -** We thank Referee 3 for the attention. As part of the complete revision of the manuscript, this table was omitted. The sensors we used are now described in the last paragraph of the 'Materials and Methods' section.

**SC 5 -** *Table 1 and Table 2 must be reversed.*
**Autors' response -** As mentioned above, according to Referees' comments, Table 2 (that originally described sensors' position) was omitted. Following Table 1, we now

present the TWW composition.

**SC 6 -** *Lines 95-97: the sentence should be rewritten to be clearer.*
**Autors' response -** We accept the comment and made improvements accordingly: "Glucose was chosen as the main carbon source for two reasons: in addition to the fact that it is often used in synthetic WW for laboratory SAT systems (Essandoh et al., 2011; Ak et al., 2013), its high consumption rate by bacteria (compared to more complex carbohydrates or humic material) allowed the investigation of the system's behavior around the ranges of ORP values that are found in field SAT systems (Orgad et al., 2017)."

**SC 7 -** *Line 100/Table 1: why call the inflow of experiments 3 and 4 "Real TWW" while additions of glucose and ammonium have been made? If the explanation comes later, put it here.*
**Autors' response -** The TWW for the third and fourth experiments were collected from the Dresden WWTP after an activated sludge process. This means that the microbial community present in the TWW itself was inherently different than the synthetic WW (that were prepared with tap water). The addition of glucose and $NH_4^+$ was necessary in order to equalize the inflow DOC, TKN and $NH_4^+$ concentrations between all four experiments. A more precise term would be "ammended real wastewater", but that would be cumbersome. We did clarify the terminology in the sentence, which reads now: " The real TWW used for experiments RW150 and RW240 were enriched with glucose and $NH_4^+$ after initial chemical analysis (presented in the supplementary material) to match the $NH_4^+$, TKN and DOC concentrations to these of the synthetic TWW."

**SC 8 -** *Table 1: in experiment 3, in the line "inflow" it misses the letter "T" because it is treated wastewater that was added and not raw wastewater.*
**Autors' response -** We thank Referee 3 for the attention. Corrected according to

comment.

**SC 9 -** *Line 102: the sentence starting with "During all experiment, ..." should be the beginning of a new paragraph because it concerns ALL the expermientations and not only the experiment 3 and 4. Refer to Table 2. By the way, it lacks an S to "experiment".*
**Autors' response -** As part of this section's revision, we moved this line to the last paragraph of the 'Materials and Methods' section (that describes the sensors). It is now in a separate paragraph as was suggested. Typo was corrected according to comment.

**SC 10 -** *Lines 112-114: the first sentence has already been mentioned above (line 100) and the second sentence should be after line 100.*
**Autors' response -** This line was improved: "The real TWW used for experiments RW150 and RW240 were enriched with glucose and $NH_4^+$ after initial chemical analysis. (presented in the supplementary material) to match the $NH_4^+$, TKN and DOC concentrations to these of the synthetic TWW. Final $NH_4^+ - N$, TKN and DOC concentrations for the synthetic and real WW are resented in Table 2". However, we think the second part of the sentence, referring to the enrichment of the TWW belongs in this line (and not in line 100) since we believe this information should appear after the description of the synthetic WW composition.

**SC 11 -** *Lines 121: remove the ":" which would indicate a list behind whereas here the different compounds and their methods of determination are separated by dots.*
**Autors' response -** We fully accept the comment. The four methods used are now separated by ';'.

**SC 12 -** *Line 121: why do you write "ammonium" and not NH4+ whereas it has already been defined line 85? True for the whole document.*

[Figure]

**Autors' response -** We accept that consistent use of the chemical formula of ammonium ($NH_4^+$) is preferable. Hence, we now use it throughout the manuscript.

**SC 13 -** *Line 122: it misses the sign "-" behind NO2.*
**Autors' response -** We thank Referee 3 for the attention. Corrected according to comment.

**Results and Discussion**

**SC 14 -** *Lines 140-142: repetition of Haaken et al., 2016*
**Autors' response -** Corrected according to comment.

**SC 15 -** *Line 169: you say ∼50 minutes on average for part 1 whereas you said line 132 ∼ 80 minutes. Be consistent.*
**Autors' response -** We thank Referee 3 for the attention. This error was corrected.

**SC 16 -** *Line 179: 3 digits after the decimal point for the minutes are not necessary (2.7 minutes instead of 2.700 minutes).*
**Autors' response -** In this line, the commas (e.g in 2,700) do not symbolize a decimal points but thousands separators.

**SC 17 -** *Line 186: 'around' is not necessary because you write "∼ ". Moreover, write "for the 375 cm sensor" and "for the 575 cm sensor" instead of "in the 375 cm sensor" and "in the 575 cm sensor".*
**Autors' response -** The word 'around' was omitted as suggested. However, we do not believe the word 'for' is suitable for the purpose of this sentence.

**SC 18 -** *Lines 194-195: again, this information has already be written line 100.*
**Autors' response -** As this is the first time in the results and discussions section that

data with real TWW is presented, we think it is important to remind the difference between these experiments and the former ones. However, we accept the comment and the sentence, that now reads " In these experiments we used real TWW" was shortened.

**SC 19 -** *Figure 4: it would be better to display the input concentrations on the graphs to better see the differences between input and output for experiments 3 and 4.*
**Autors' response -** As mentioned in the text, input parameters (DOC, TKN and $NH_4^+$) were the same for both experiments (inflow concentrations are presented in Table 2 in the main text). Since the aim of this figure is to show the difference between the two experiments, we believe that addition of the input concentrations will add unnecessary complexity to the figure.

**SC 20 -** *Lines 203 and 205: the numbers in the parentheses are the differences between the concentrations measured at the input and those measured at the output for the experiments 3 and 4? I think that it is not wise to express the efficient removal in terms of differences in concentrations but you should rather express these removal efficiencies in terms of percentage.*
**Autors' response -** The numbers in parentheses represent outflow concentrations. We accept that this is not clear from the sentence and hence we improved its structure: "Outflow $NH_4^+ - N$, TKN and DOC concentrations during RW240 ($\sim$ 0.033, $\sim$ 0.62 and $\sim$ 1.65 mg/L respectively) were significantly lower compared to their inflow concentrations . During RW150, $NH_4^+ - N$, TKN and DOC outflow concentrations ($\sim$ 0.5, $\sim$ 3.8 and $\sim$ 4.4 mg/L, respectively) were also lower compared to the inflow, but averaged significantly higher compared to RW240 (t-test, $\alpha$=0.05) ..

**SC 21 -** *Line 203: you say that your measurements correspond to what is measured in the full scale SAT site but we have no table, figure, or at least a reference on which your statement is based.*

**Autors' response -** Figure 5 was designed specifically to demonstrate this claim. The data presented in Figure 5a is based on field observations from one of the SHAFDAN's infiltration ponds, as explained in detail in the 'Comparison with field observations' section.

**SC 22 -** *Line 253: Table 1 should be Table 2.*
**Autors' response -** As was mentioned before, the original Table 2 was omitted.

**Summary and Conclusions**

**SC 23 -** *Line 280: 150 minutes or 240 minutes (and not only 240 m which means meter).*
**Autors' response -** We thank Referee 3 for the attention. Corrected according to comment.